# Biological Traits and Comprehensive Genomic Analysis of Novel *Enterococcus faecalis* Bacteriophage EFP6

**DOI:** 10.3390/microorganisms12061202

**Published:** 2024-06-14

**Authors:** Sajjad Ahmad, Qingwen Leng, Gongmingzhu Hou, Yan Liang, Yanfang Li, Yonggang Qu

**Affiliations:** College of Animal Science and Technology, Shihezi University, Shihezi 832003, China; ahmadsajjad@stu.shzu.edu.cn (S.A.); qingwenleng@shzu.edu.cn (Q.L.); hougongmingzhu@foxmail.com (G.H.); liangyan@shzu.edu.cn (Y.L.); yanfangli@shzu.edu.cn (Y.L.)

**Keywords:** *E. faecalis*, phage therapy, embryonic mortality, genome sequencing

## Abstract

*Enterococcus faecalis* is a prevalent opportunistic pathogen associated with chicken embryonic and neonatal chick mortality, posing a significant challenge in poultry farming. In the current study, *E. faecalis* strain EF6, isolated from a recent hatchery outbreak, served as the host bacterium for the isolation of a novel phage EFP6, capable of lysing *E. faecalis*. Transmission electron microscopy revealed a hexagonal head and a short tail, classifying EFP6 as a member of the *Autographiviridae* family. EFP6 showed sensitivity to ultraviolet radiation and resistance to chloroform. The lytic cycle duration of EFP6 was determined to be 50 min, highlighting its efficacy in host eradication. With an optimal multiplicity of infection of 0.001, EFP6 exhibited a narrow lysis spectrum and strong specificity towards host strains. Additionally, EFP6 demonstrated optimal growth conditions at 40 °C and pH 8.0. Whole genome sequencing unveiled a genome length of 18,147 bp, characterized by a GC concentration of 33.21% and comprising 25 open reading frames. Comparative genomic assessment underscored its collinearity with related phages, notably devoid of lysogenic genes, thus ensuring genetic stability. This in-depth characterization forms the basis for understanding the biological attributes of EFP6 and its potential utilization in phage therapy, offering promising prospects for mitigating *E. faecalis*-associated poultry infections.

## 1. Introduction

The poultry industry stands as one of agriculture’s most lucrative and promising sectors across numerous countries globally [1,2], furnishing high-quality dietary staples, particularly meat [3,4]. Despite its profitability, intensive broiler chicken rearing practices have adverse effects on their health, productivity, and safety [5,6], resulting in substantial economic losses [7,8]. An avenue to enhance poultry production while offering organic food options involves leveraging probiotics [9,10]. These formulations comprise live bacterial cells with beneficial effects on the host organism [11,12]. Among the microorganisms in probiotics, a significant portion includes lactic acid-producing strains, such as *Enterococci* Take *E. faecalis* Symbioflor 1 (SymbioPharm, Herborn, Germany), for example, which is used to treat recurrent upper respiratory tract illnesses in humans as well as to prevent and treat diarrhea in pigs, poultry, livestock, and pets. *E. faecium* SF68^®^ (NCIMB 10415; Cerbios-Pharma SA, Barbengo, Switzerland) is widely used as a medicine in humans and as a feed additive for a variety of animals [13,14,15]. On the other hand, studies have reported *E. faecalis* as an opportunistic pathogen that can affect poultry production, according to [16]. Certain sequence types (STs) of *E. faecalis* are linked to specific conditions in poultry, such as amyloid atrophy (AA), first-week mortality, septicemia, and salpingitis. ST82, ST174, and ST177 are prominent in mortality among broiler breeders and may have a zoonotic connection, as some types are associated with human infections [17].

Furthermore, *Enterococcus*-associated yolk sac infections have become significant in the poultry industry, with *E. faecalis* and *E. cecorum* being economically impactful pathogens [18,19,20,21]. *E. hirae* and *E. durans* have been linked to severe conditions such as encephalomalacia and endocarditis [22]. *Enterococci* are also recognized as opportunistic pathogens in humans, causing various nosocomial infections, with *E. faecalis* and *E. faecium* being the predominant species [23,24]. Additionally, in poly-microbial infections, synergistic interactions among microorganisms, as observed in wound infections, contribute to disease pathogenesis [25]. 

The escalation of antibiotic resistance in *E. faecalis* due to the inappropriate use of antibiotics in clinical settings leads to the rise in multidrug-resistance bacteria within adverse environments, particularly *Vancomycin*-resistant *Enterococci*, which present significant therapeutic challenges, especially in immuno-compromised patients [24,26,27]. The existence of biofilms leads to increased tolerance, drug resistance, and mutagenicity of *E. faecalis* [28]. Therefore, it is essential to establish a reliable and efficient approach to counter *E. faecalis* in hatcheries and poultry facilities, aiming to prevent infections and enhance the efficacy and safety of these operations. Bacteriophages, often called phages, are viral agents that selectively target and eradicate pathogenic bacteria by triggering their lysis. Hence, phage therapy exclusively utilizes lytic phages. This therapeutic approach has shown favorable outcomes in treating various diseases; phage therapy has found application in both the agricultural and food sectors, with recent utilization for biological control purposes in the United States food industry [29,30,31]. Promising results were achieved through the utilization of an actinobacteriophage for periodontitis treatment [32]. Another investigation involved the establishment of an ex vivo model simulating *E. faecalis*-associated periapical periodontitis, where the *E. faecalis* phage (EFDG1) effectively suppressed the formation of *E. faecalis* biofilms [33]. 

The current study emphasizes the isolation of a novel lytic bacteriophage, EFP6, exhibiting specific activity against multidrug resistance (Kanamycin, Gentamicin, Streptomycin, Nalidixic Acid, Erythromycin, Tetracycline, and Rifampicin) and highly virulent (agg, gelE, fsr, CyIA, AsaI, Asa373, Ace, and EF0591)-positive *E. faecalis* strains (EF6 and EF5). The isolated phage was subjected to a characterization of its biological features and whole genome analysis.

## 2. Materials and Methods

### 2.1. Bacterial Cultures and Culture Media

The multidrug-resistant and virulent strains of *E. faecalis* (EF5 and EF6), which were isolated from a recent hatchery outbreak, were preserved in the New Veterinary Drug Development Laboratory at Shihezi University, Xinjiang, China. For host range, a laboratory collection of 45 bacterial strains was sourced from chickens. These included *Barvi bacterium*, *Acidovorox*, *Proteus mirabilis*, *Staphylococcus aureus*, *Staphylococcus argenteus*, *Klebsiella pneumonia*, *Enterococcus* spp., *Acinobacter baumannii*, *E. coli*, and *E. faecalis*. The strains were cultured overnight either on Brain Heart Infusion Agar (Solarbio Science and Technology Co., Ltd. Beijing, China.) or in Brain Heart Infusion Liquid culture (Solarbio Science and Technology Co., Ltd. Beijing, China) at 37 °C.

### 2.2. Isolation, Purification, and Initial Screening of the Lytic Bacteriophage

The bacteriophage was isolated from waste-water in Shihezi using a modified protocol from [34]. Waste-water from poultry was mixed with 5 × BHI and bacterial strains subsequently incubated overnight at 37 °C. After centrifugation and filtration (0.22 μm), a phage lysate was obtained. The presence of the phage was determined by spot tests on *E. faecalis* strain EF6. Preliminary screening was performed using the double-layer agar (DLA) method [35], with the diluted phage lysate and incubated with host bacteria. Plaque formation on solid agar plates indicated successful phage purification after 3 rounds. Phage plaques were examined for morphology, and titer was determined. Before being used in a later experiment, the resulting phage stock was kept at 4 °C.

### 2.3. Assessment of Bacteriophage Host Spectrum

The host spectrum was determined following the methodology of [36]. A panel of 45 bacterial strains, including *E. faecalis* (10 strains), *A. baumannii* (2 strains), *Acidovorax* (1 strain), *B. bacterium* (1 strain), *E. coli* (12 strains), *K. pneumonia* (2 strains), *P. merrabullis* (8 strains), *S. aurus* (2 strains), and 6 strains of *S. argenteus* (Table 1), was utilized. In total, 5 μL of a pure phage lysate of EFP6 (7.7 × 10^7^ plaque-forming units, PFUs) was applied onto a BHI solid medium spread with the respective bacterial strains. Using sterile distilled water as a control, the bacterial cultures were incubated at 37 °C for 7 to 12 h. The development of distinct plaques verified the bacteria’s susceptibility to EFP6. Plaque-positive (+) and plaque-negative (−) groups were created based on the results.

### 2.4. Biological Traits of EFP6

Four milliliters of a bacteriophage supernatant were divided into four separate 1.5 mL EP tubes and exposed to various temperatures (40 °C, 50 °C, 60 °C, 70 °C) in a water bath for durations of 10, 20, 30, 40, 50, and 60 min. At each time interval, 100 μL of the bacteriophage supernatant was obtained, and the phage titer was determined using the double-layer agar plate method. The impact of temperature on bacteriophage titer was graphed using Graph Pad Prism version 8.0.2. The experiment was repeated three times.

The liquid medium using Brain Heart Infusion (BHI) was made to have a pH from 3.0 to 11.0. Next, 900 µL of BHI was mixed with 100 µL of the phage solution that contained 7.7 × 10^7^ plaque-forming units per milliliter (PFU/mL). The DLA method was used to determine the phage titer after a 37 °C incubation period of one hour [37].

For UV light treatment, the bacteriophage supernatant underwent continuous UV irradiation for 2 h, with phage titer assessed at 20 min intervals [38].

EFP6 was combined with chloroform at 0%, 1%, 2%, 4%, and 5% concentrations. Then, there was an hour of incubation at 37 °C and 180 rpm of shaking. The DLA approach was utilized to ascertain the phage titer [39].

The host bacteria and bacteriophage EFP6 were combined at various multiplicities of infection (MOIs) ranging from 0.0001 to 10, and the mixture was then cultivated for 4 h at 37 °C with 180 rpm shaking. A 0.22 µm filter was used to filter the supernatant after it was centrifuged for 10 min at 12,000× *g* [40]. Using the DLA approach, the phage titer of several MOIs from samples was ascertained.

*E. faecalis* (EF6) was cultured in a BHI liquid medium until OD600nm reached 0.2. The bacteriophage (10^7^ PFU/mL) was added, and growth was monitored hourly. A growth curve of host bacteria was plotted. The experiment was replicated thrice [41].

### 2.5. One-Step Growth Curve of EFP6

We adapted the methodology outlined in [38]. In brief, bacteriophage EFP6 was utilized to infect host bacteria EF6 (with an OD600 value of 0.5). The absorption process was conducted at 37 °C for 10 min; unbound bacteriophages were discarded by centrifugation and the pellets were re-suspended in 5 mL of a sterile BHI medium. Samples were collected at 9 different time intervals (0 min, 10 min, 20 min, 30 min, 40 min, 50 min, 60 min, 90 min, and 120 min) followed by centrifugation at 12,000 rpm for 10 min at 4 °C. Subsequently, the bacteriophage titer was promptly determined using the double-layer agar plate method. Each experiment was conducted with three replicates. The ratio of the average phage titer value during the plateau phase to the average titer value of the incubation period was used to determine the phage burst size.

### 2.6. Concentration and Electron Microscopic Observation of EFP6

To concentrate the bacteriophage, a protocol was adopted from [42] with slight adjustments. Briefly, an overnight culture of the bacteriophage lysate (600 mL) was centrifuged at 8000 rpm for 15 min at 4 °C. The resulting supernatant was filtered through a 0.22 μm membrane and treated with DNase I and RNase A. After incubation, NaCl was added to achieve a final concentration of 1 M, followed by centrifugation at 12,000 rpm for 20 min at 4 °C. PEG 8000 was dissolved in the supernatant, and the mixture was incubated at 4 °C for three hours. After centrifugation, the supernatant was discarded, and the phage particles were suspended in an SM buffer. Chloroform was added, and the mixture was centrifuged three times before being stored at 4 °C for later use. For electron microscopic observation, a drop of the bacteriophage concentrate was placed on a dry slide, and a copper grid was placed on top. After interaction, excess liquid was absorbed, and negative staining was performed using 2% phosphor tungstic acid. A transmission electron microscope was used to study the bacteriophage’s shape and size.

### 2.7. Whole Genome Sequencing of Phage EFP6

The concentrated bacteriophage was sent to Shenzhen Huitong Biotechnology Co., Ltd., China, who conducted the library construction, sequencing, and data quality control for phage EFP6. The sequencing platform employed was Illumina NovaSeq www.illumina.com, accessed on 15 December 2023, and assembly was performed using Newbler software (Version 2.9.0). A comparative analysis of the phage’s complete genome sequence was carried out using NCBI BLAST+ software (Version 2.13.0) [43].

The size, GC content, and nucleotide composition of the EFP6 genome were analyzed using EditSeq software (Version 11.1.0.54). The presence of antibiotic-resistance genes in the EFP6 genome was investigated using the CARD Antibiotic Resistance Gene Database (https://card.mcmaster.ca/, accessed on 1 April 2024), while the VFDB Virulence Factor Database (http://www.mgc.ac.cn/VFs/, accessed on 2 April 2024) was utilized to identify virulence factor genes. A circular map illustrating sequence features, nucleotide composition, and GC skew of the entire phage genome was generated using the genome utilization online tool accessible at https://js.cgview.ca/ (accessed on 10 April 2024) and Gene marks (http://exon.gatech.edu/GeneMark/genemarks.cgi, accessed on 10 April 2024) [44]; the prediction of protein-coding genes within the phage EFP6 genome was conducted. The predicted functions were validated using the ORF Finder provided by NCBI (https://www.ncbi.nlm.nih.gov/orffinder/, accessed on 5 March 2024), and genes were annotated based on an E-value threshold lower than 10^−3^. The identification of tRNAs within the genome was carried out using tRNAscan-SE software (http://lowelab.ucsc.edu//tRNAscan-SE/, accessed on 5 March 2024), with a cove score threshold set to 20.

### 2.8. Phylogenetic Analysis and Collinearity of Phage EFP6

The phylogenetic analysis of EFP6 was conducted, using DNA polymerase (ORF8), lysin (ORF14), tail protein (ORF18), and capsid protein (ORF20) [28]. Protein sequences were initially compared using the BLAST tool on the NCBI website, and sequences from phages with high homology were selected. A phylogenetic tree was then constructed using Mega 11.0 software [45].

Phage genomes exhibiting the highest similarity to EFP6 were retrieved from the NCBI database for the collinearity analysis. Easy Fig software (Version 2.2.5) was employed to generate linear comparison maps of multiple genomic loci based on BLAST alignments [45]. Additionally, VIRIDIC, an online tool accessible at http://rhea.icbm.uni-oldenburg.de/VIRIDIC/, accessed on 10 April 2024, was utilized for virus genome distance calculation [42]. This server employs traditional algorithms endorsed by the International Committee on Taxonomy of Viruses (ICTV) to analyze VIRIDIC data from various phages, including the short-tailed *Enterococcus* phage.

## 3. Results

### 3.1. Isolation, Purification, and Preliminary Screening of the Lytic Bacteriophage

Following three rounds of purification, all EFP6 showed clear plaques. The plaque diameters were recorded as 5 mm ± 0.5 mm (Figure 1A). The observation of bacteriophage EFP6 under transmission electron microscopy shows that its head exhibits a hexagonal shape, with a head diameter of approximately (57 ± 5) nm and a relatively short tail (Figure 1B). According to the classification and naming guidelines set forth by the International Committee on Taxonomy of Viruses, it is classified under the family *Autographiviridae*, and it is designated as vB_EFS_EFP6, where “vB” denotes virus bacteriophage and “EFS” represents *E. faecalis*.

### 3.2. Determination of Bacteriophage Host Range

EFP6 was tested against various bacterial species, including *B. bacterium* (1 strain), *Acidovorox* (1 strain), *A. baumannii* (2 strains), *P. mirabilis* (8 strains), *S. aureus* (2 strains), *S. argenteus* (6 strains), *K. pneumoniae* (2 strains), *E. coli* (12 strains), and *E. faecalis* (10 strains). The results showed that EFP6 has a strong narrow host range, lysing only four strains of *E. faecalis*. All other bacterial species as well as the remaining strains of *E. faecalis* exhibited resistance to EFP6 (Table 1).

### 3.3. Biological Characteristics of EFP6

Analyzing bacteriophage EFP6 under various temperatures showed that it is consistently highly potent at 40 °C for 40 min and slightly increased after 40 to 60 min; however, at 50 °C and after 20 min, the potency slightly dropped, and a sudden drop down was recorded at 60 °C, and was quickly deactivated at 70 °C (Figure 2A).

Under varying pH values (from 3 to 11), the infectivity of EFP6 showed that it was stable between 4.0 and 10.0 pH. However, at pH 6.0, the largest titer was recorded (Figure 2B).

EFP6 showed susceptibility to UV radiation. Under UV irradiation, the potency of EFP6 declined gradually between 0 and 100 min while after 120 min, a notable reduction in its potency was recorded; overall, the potency decreased from log10 9.6 PFU/mL to log10 8.6 PFU/mL (Figure 2C).

After exposure to various concentrations of chloroform following an hour of incubation, potency measurements indicate slight reductions compared to the untreated control. Specifically, potency decreases by 0.23, 0.26, 0.36, and 0.47 log PFU/mL with chloroform concentrations of 1, 2, 4, and 5%, respectively (Figure 2D). These findings suggest that the bacteriophage exhibits low sensitivity to chloroform, rendering it suitable for incorporation into large-scale preparations involving chloroform.

When the MOI (multiplicity of infection) is set to 0.001, the potency of bacteriophage EFP6 peaks at 9.77 log10 PFU/mL, suggesting that the optimal infection multiplicity for bacteriophage EFP6 is 0.001; the finding is illustrated in Figure 2E.

Utilizing the EF6 strain as the test subject, experimental results demonstrate that bacteriophage EFP6 exhibits significant inhibitory effects on the growth of its host bacteria; specifically, its host strain showed complete inhibition for 6 h (Figure 2F).

### 3.4. One-Step Growth Curve of EFP6

The findings are depicted in Figure 3. The efficacy of bacteriophage EFP6 notably rises after 10 min and reaches a stable level by 60 min, exhibiting a brief latency period. During the first 30 min, the potency of the bacteriophage sharply increases, peaking at 9.76 log10 PFU/mL during the burst phase. Utilizing calculations derived from the formula, the lysing capacity of bacteriophage EFP6 is estimated to be approximately 127 PFU/cell.

### 3.5. Whole Genome Analysis of Phage EFP6

The genome of phage EFP6 spans 18,147 base pairs, with a composition of adenine (A) at 33.96%, thymine (T) at 32.83%, cytosine (C) at 16.08%, and guanine (G) at 17.13%, resulting in a GC content of 33.21% (Appendix A). It comprises 25 open reading frames (ORFs), with 15 ORFs assigned functions and the remaining 10 designated as hypothetical proteins. No tRNA genes were identified within the genome. A BLAST homology analysis classifies phage EFP6 within the order *Caudovirales* and the *Autographiviridae* family, specifically as a member of short-tailed bacteriophages.

The genome map of phage EFP6, depicted in Figure 4, was generated using CGView. Triangular arrows in different colors denote ORFs, with orange indicating structural and packaging proteins, purple representing metabolism-related proteins, blue depicting DNA replication and regulatory proteins, and yellow marking hypothetical proteins. Most genes are located on the positive strand. The middle circle illustrates GC skew, while the innermost circle displays the GC content.

### 3.6. Functional Gene Analysis of Bacteriophage EFP6

The functional annotation of phage EFP6 genes reveals a diverse array of functional modules, including lysis, metabolism, structure and packaging, and DNA replication. Out of the 25 identified genes, 11 exhibit significant homology to known functional proteins, while the remaining 14 encode hypothetical proteins. Notably, no lysogenic, antibiotic resistance, or virulence genes were detected (Table 2, Appendix A). Structural Proteins: EFP6 contains six structural proteins, including putative encapsulation protein (ORF7), tail structure proteins (ORF18), minor capsid protein (ORF20), lower collar and collar proteins (ORF21 and 22), and putative major head protein (ORF23). Metabolism-Related Proteins: EFP6 possesses three metabolism-related proteins, including an APC amino acid–polyamine–organocation transporter (ORF11), HNH homing endonuclease (ORF13), and negative regulator of beta-lactamase expression (ORF16). Lysis: This category includes CHAP domain-containing phage lysin (ORF14) and a negative regulator of beta-lactamase expression (ORF16). DNA Replication and Regulation: EFP6 encompasses two genes in this category, namely standard DNA binding protein (ORF3) and DNA polymerase (ORF8). The need for novel antimicrobial treatments has sparked renewed attention in phage therapy. To ensure safety, it is imperative to avoid phages harboring genes linked to antibiotic resistance, lysogeny, and virulence when considering them as biocontrol agents [28]. Hence, phage EFP6 demonstrates promise as an antimicrobial agent. The comprehensive functional analysis sheds light on the genetic repertoire of phage EFP6, providing insights into its potential mechanisms of action in bacterial lysis and replication.

### 3.7. Evolutionary Analysis of Bacteriophage EFP6

The DNA polymerase (ORF8), lysin (ORF14), tail protein (ORF18), and capsid protein (ORF20) of phage EFP6 are closely related to those of E. phage EfaPEf 6.2 (NC 049932.1), E. phage Efae 230P4 (NC 025467.1), and E. phage N3 (ON352054.1), respectively, according to an analysis of the evolutionary relationships between phage EFP6 and other phages (Figure 5A–D). The reason for this resemblance could be that when many phages infect the same host cell, mobile genes are transferred during proliferation. This leads to the accumulation of mutations in common gene sequences between phages that target the same disease. Since other phages targeting *Enterococcus* spp. developed different clusters, phage EFP6 clustered with *E. faecalis* phages E. phage Efae 230P4 and E. phage N3.

### 3.8. Collinearity Analysis

Using DNA polymerase, lysis, tail protein, capsid protein, and *E. faecalis* phages as bases for the phylogenetic analysis, phage EFP6’s complete genome exhibited high similarity with E. phage EfaPEf 6.2 (NC 049932.1), E. phage Efae 230P4 (NC 025467.1), and E. phage N13 (ON352054.1). As a result, phage EFP6 was selected for collinearity investigation with various other phages, and it shows high collinearity ranging from 100 to 68% (Figure 6).

## 4. Discussion

While the majority of *E. faecalis* phages recovered by researchers belong to the *Siphoviridae* family, with a few exceptions such as vB_EfaP_IME195, transmission electron microscopy demonstrated that the morphology of phage EFP6 is compatible with the *Autographiviridae* family, matching with findings from prior studies [46,47]. Thus, EFP6 represents the first reported *E. faecalis Autographiviridae* phage in China. EFP6 exhibited initial deactivation at a temperature of 60 °C and pH 11.0, which contradicts the findings obtained from E. phage Efae 230p4 (NC025467.1) [48]. However, both studies showed resemblance in terms of chloroform tolerance; similarly, EFP6 showed a difference in its lytic cycle and the measurement of its head with *Enterococcus* phage N13(ON352054). EFP6 exhibits a limited lysis spectrum, targeting only four strains of *E. faecalis* (*E. faecalis* 5, 6, 7, and 8) while demonstrating no action against the investigated strains of *P. merrabulus*, *S. aureus*, *E. coli*, and other *E. facealis*. This selectivity shows a potent capacity to target pathogens while protecting beneficial strains of *E. faecalis*. Broad phage lysis capabilities might not be beneficial because not all *E. faecalis* strains are harmful and some are probiotics used in clinical treatment. While EFP6’s complete genome has a wealth of genetic material and a variety of activities, it is comparatively smaller than the majority of published *E. faecalis* phage genomes. Important genetic information can be found throughout an organism’s whole genome. The genomic sequence of the recently isolated EFP6 is highly similar to the genomic sequences of *E. faecalis* phage N13 and vB_Efap_efmus4, despite the fact that the genetic characterization of the genomes of several *E. faecalis* bacteriophages has been established in earlier works. The genomic sequence lengths of the three *E. faecalis* phages (18,147, 18,449, and 18,186 bp) are comparable [49,50]. The three *E. faecalis* phages can be grouped together in the same subclade based on their capsid protein (Figure 5). The collinearity of the whole genome between EFP6 and E. phage EfaPEf 6.2 (NC 049932.1), E. phage Efae 230P4 (NC 025467.1), and E. phage N3 (ON352054.1) is 68% to 100%. Two proteins involved in the degradation of bacterial cell walls were predicted (ORF 14 and ORF 17) to be phage proteins for typical two-component bacterial cell lysis. ORF3, ORF7, and ORF 23 encapsidation protein; putative single-stranded DNA binding protein; and putative major head protein are 98.18%, 97.82%, and 95.26% homologous to *Enterococcus* phage vB Efae230P-4 while ORF8 DNA polymerase, ORF13 HNH homing endonuclease, ORF 14 CHAP domain-containing phage lysin, ORF16 negative regulator of beta-lactamase expression, ORF 17 Holin, ORF 20 minor capsid protein, ORF21 lower collar protein, and ORF22 collar protein are 97.44%, 93.41%, 97.46%, 95.72%, 99.23%, 91.17%, 97.75%, and 95.38% homologous to the *Enterococcus* phage vB EfaP IME195, respectively. Holin is a protein that perforates the membrane and inserts into the bacterial cell. The CHAP domain-containing phage lysin is a type of enzyme produced by EFP6 to facilitate the release of newly formed phage particles from infected bacterial cells. The CHAP domain is a conserved protein domain found in certain bacterial and phage proteins. It is named after the cysteine, histidine-dependent amidohydrolase/peptidase motif it contains [51]. Proteins with CHAP domains often exhibit amidase or peptidase activity, meaning that they can cleave specific peptide bonds in proteins or peptidoglycan, a component of bacterial cell walls. Phage lysins, also known as endolysins, are enzymes produced by EFP6 during the late stages of the viral replication cycle. These enzymes target and degrade specific components of the bacterial cell wall, causing lysis of the cell membrane and releasing the newly formed phage particles. Lysins are essential for the release of phage progeny from the host bacterium, enabling the spread of the viral infection to neighboring bacteria. Therefore, a CHAP domain-containing phage lysin is an enzyme produced by EFP6 that contains a CHAP domain and is involved in breaking down bacterial cell walls, aiding in the release of viral progeny. These enzymes have garnered interest for their potential applications in bacteriophage therapy, a promising approach for combating antibiotic-resistant bacterial infections.

Future research will focus on exploring the therapeutic potential of EFP6 for bacterial infections and investigating its application in developing and harnessing phage therapy for embryo sac infections and neonatal chick mortality.

## 5. Conclusions

Bacteriophage EFP6, isolated from waste-water using EF6 as host bacteria, has a narrow host range. It exhibits an infective dose of 0.001 and achieves an average lysis of approximately 127 PFU per cell. EFP6 demonstrates adaptability to various temperatures and pH ranges, with sensitivity to UV radiation, and resilience to chloroform. The linear double-stranded DNA spans 18,147 bp with a GC content of 33.12% and lacks tRNA, lysogenic genes, known antibiotic-resistance genes, and virulence genes. The genome of EFP6 contains several key proteins, including two involved in bacterial cell wall degradation (ORF 14 and ORF 17), which are essential for the EFP6 lytic cycle. High homology was observed between these proteins and those in other *Enterococcus* phages. The presence of a CHAP domain-containing phage lysin in EFP6 underscores its potential in bacteriophage therapy, particularly in targeting antibiotic resistance.

## Figures and Tables

**Figure 1 microorganisms-12-01202-f001:**
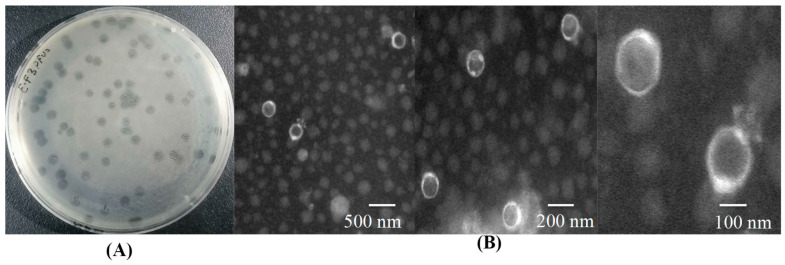
EFP6 screening. (**A**) Plaque characteristics and (**B**) morphology observed via transmission electron microscopy. Scale bar indicates 500, 200, and 100 nm, respectively.

**Figure 2 microorganisms-12-01202-f002:**
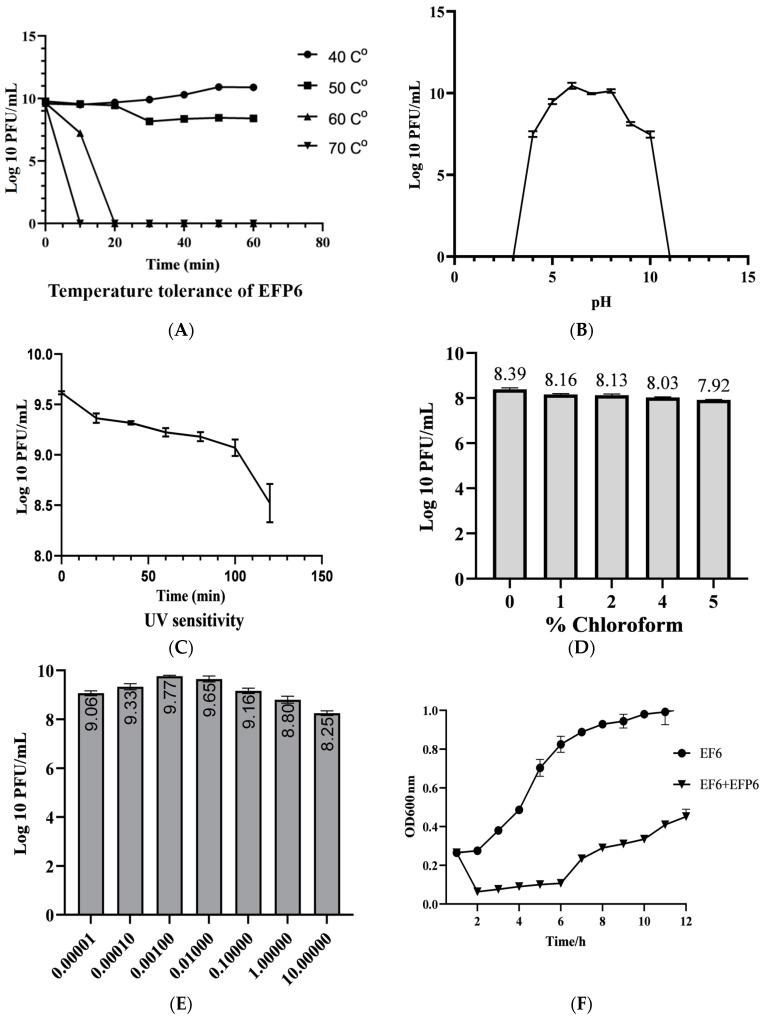
Phage EFP6 physiochemical influences. (**A**) Phage EFP6 temperature stability. (**B**) Phage EFP6 stability at varying pH values. (**C**) Sensitivity of EFP6 to UV radiation. (**D**) Effect of various percentages of chloroform on EFP6. (**E**) Optimal multiplicity of infection of bacteriophage EFP6. (**F**) Bacteriostatic activity of EFP6.

**Figure 3 microorganisms-12-01202-f003:**
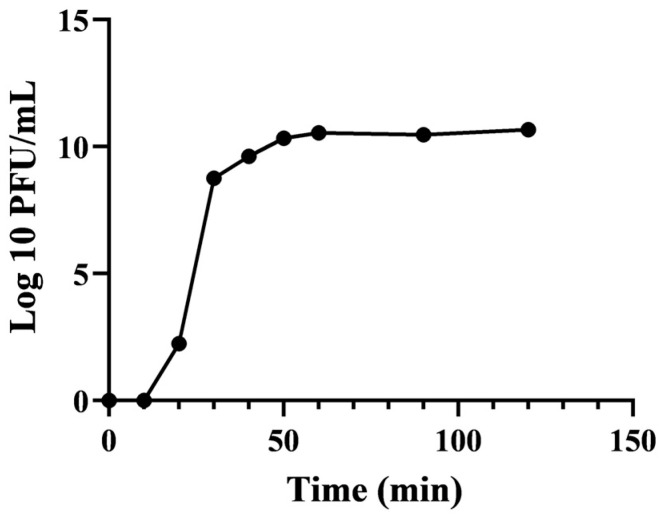
One-step growth curve of bacteriophage EFP6.

**Figure 4 microorganisms-12-01202-f004:**
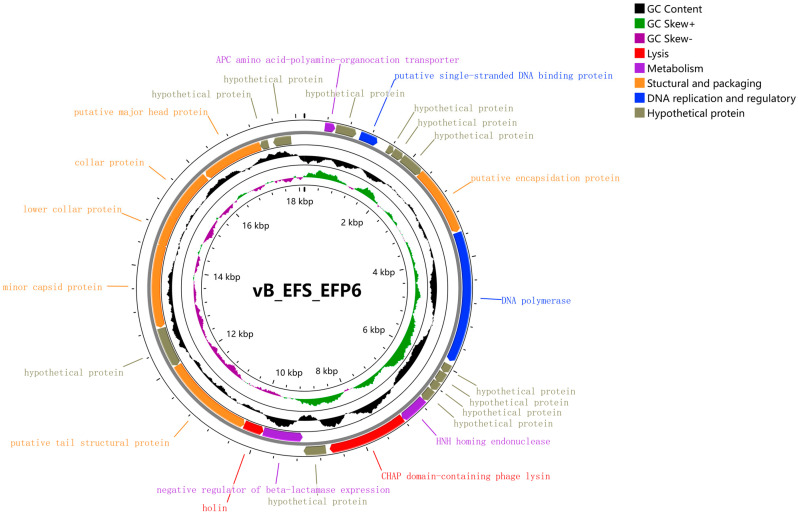
Whole genome map of bacteriophage EFP6.

**Figure 5 microorganisms-12-01202-f005:**
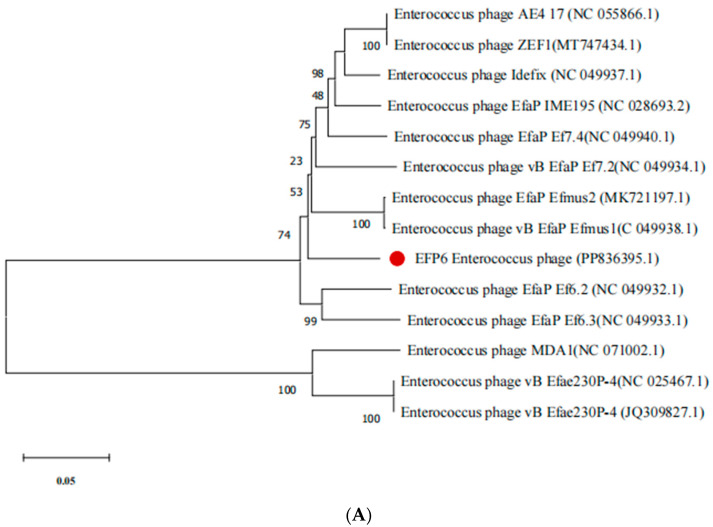
Phylogenomics of phage EFP6 based on (**A**) DNA polymerase, (**B**) lysis, (**C**) tail protein, and (**D**) capsid protein.

**Figure 6 microorganisms-12-01202-f006:**
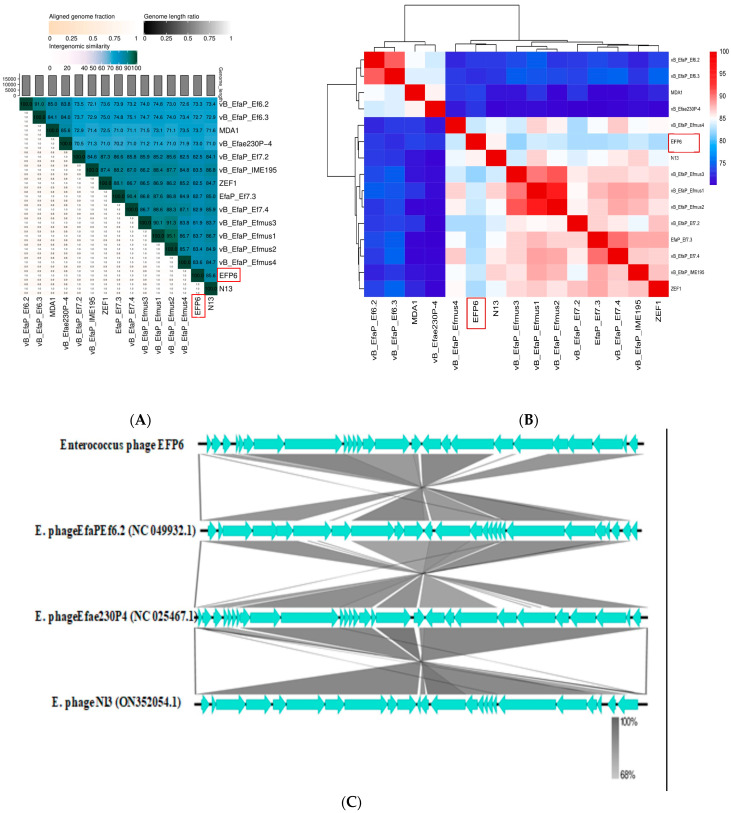
EFP6 bacteriophage homology; (**A**) percent similarity of phage EFP6 with other closely related *Enterococcus* phages, (**B**) heat map based on similarity percentage, (**C**) genome sequence comparison of EFP6 with closely related phages.

**Table 1 microorganisms-12-01202-t001:** The host spectrum of phage EFP6.

NO.	Bacteria	Source	Phage Sensitivity	NO.	Bacteria	Source	Phage Sensitivity
1	*Acidovorox*	Dead embryo yolk	Resistant	24	*E. coli* 1	Chicken liver	Resistant
2	*B. bactirium*	Dead embryo yolk	Resistant	25	*E. coli* 2	Chicken liver	Resistant
3	*A. baumannii* 1	Chicken liver	Resistant	26	*E. coli* 3	Chicken liver	Resistant
4	*A. baumannii* 2	Dead embryo yolk	Resistant	27	*E. coli* 4	Chicken liver	Resistant
5	*P. mirabilis* 1	Chicken blood	Resistant	28	*E. coli* 5	Chicken liver	Resistant
6	*P. mirabilis* 2	Chicken blood	Resistant	29	*E. coli* 6	Chicken liver	Resistant
7	*P. mirabilis* 3	Chicken blood	Resistant	30	*E. coli* 7	Chicken liver	Resistant
8	*P. mirabilis* 4	Dead embryo yolk	Resistant	31	*E. coli* 8	Chicken liver	Resistant
9	*P. mirabilis* 5	Dead embryo yolk	Resistant	32	*E. coli* 9	Dead embryo yolk	Resistant
10	*P. mirabilis* 6	Dead embryo yolk	Resistant	33	*E. coli* 10	Dead embryo yolk	Resistant
11	*P. mirabilis* 7	Dead embryo liver	Resistant	34	*E. coli* 11	Dead embryo yolk	Resistant
12	*P. mirabilis* 8	Dead embryo liver	Resistant	35	*E. coli* 12	Dead embryo yolk	Resistant
13	*S. aureus* 1	Chicken blood	Resistant	36	*E. faecalis* 1	Dead embryo liver	Resistant
14	*S. aureus* 2	Chicken blood	Resistant	37	*E. faecalis* 2	Chicken blood	Resistant
15	*Enterococcus* spp.	Dead embryo yolk	Resistant	38	*E. faecalis* 3	Dead embryo yolk	Resistant
16	*S. argenteus* 1	Chicken blood	Resistant	39	*E. faecalis* 4	Chicken blood	Resistant
17	*S. argenteus* 2	Chicken blood	Resistant	40	*E. faecalis* 5	Chicken blood	Susceptible
18	*S. argenteus* 3	Chicken blood	Resistant	41	*E. faecalis* 6	Chicken blood	Susceptible
19	*S. argenteus* 4	Chicken blood	Resistant	42	*E. faecalis* 7	Chicken liver	Susceptible
20	*S. argenteus* 5	Chicken blood	Resistant	43	*E. faecalis* 8	Dead embryo yolk	Susceptible
21	*S. argenteus* 6	Chicken blood	Resistant	44	*E. faecalis* 9	Chicken blood	Resistant
22	*K. pneumoniae* 1	Chicken liver	Resistant	45	*E. faecalis* 10	Chicken blood	Resistant
23	*K. pneumoniae* 2	Chicken liver	Resistant				

**Table 2 microorganisms-12-01202-t002:** Prediction of genes and functional annotation of bacteriophage EFP6.

ORF	Start	Stop	F/R	Predicted Function	Best Match Phage	E Value	% Identity	Accession Number
EFP6_1	363	551	F	APC amino acid–polyamine–organocation transporter	Enterococcus phage EFRM31	4 × 10^−41^	84.04%	NC_015270.1
EFP6_2	564	938	F	hypothetical protein	Enterococcus phage vB_EfaP_IME195	2 × 10^−25^	57.26%	YP_009191320.1
EFP6_3	1012	1344	F	putative single-stranded DNA binding protein	Enterococcus phage vB_Efae230P-4	5 × 10^−72^	98.18%	YP_009103979.1
EFP6_4	1547	1672	F	hypothetical protein	Enterococcus phage IME_EF3	1 × 10^−19^	85.85%	NC_023595.2
EFP6_5	1669	1866	F	hypothetical protein	Enterococcus phage vB_EfaP_IME195	5 × 10^−8^	39.34%	YP_009191325.1
EFP6_6	1863	2285	F	hypothetical protein	Enterococcus phage vB_EfaP_IME195	2 × 10^−97^	98.57%	YP_009191326.1
EFP6_7	2278	3519	F	putative encapsidation protein	Enterococcus phage vB_Efae230P-4	0.0	97.82%	YP_009103976.1
EFP6_8	3532	5880	F	DNA polymerase	Enterococcus phage vB_EfaP_IME195	0.0	97.44%	YP_009191328.1
EFP6_9	5941	6111	F	hypothetical protein	Enterococcus phage vB_EfaP_IME195	1 × 10^−26^	87.50%	YP_009191329.1
EFP6_10	6111	6317	F	hypothetical protein	Enterococcus phage vB_Efae230P-4	3 × 10^−20^	57.35%	YP_009103972.1
EFP6_11	6308	8481	F	hypothetical protein	Enterococcus phage vB_EfaP_IME195	2 × 10^−24^	84.21%	YP_009191331.1
EFP6_12	6482	6715	F	hypothetical protein	Enterococcus phage vB_EfaP_IME195	1 × 10^−23^	71.67%	YP_009191332.1
EFP6_13	6712	7215	F	HNH homing endonuclease	Enterococcus phage vB_EfaP_IME195	1 × 10^−112^	93.41%	YP_009191333.1
EFP6_14	7208	8626	F	CHAP domain-containing phage lysin	Enterococcus phage vB_EfaP_IME195	0.0	97.46%	YP_009191334.1
EFP6_15	8694	9092	F	hypothetical protein	Enterococcus phage vB_EfaP_IME195	4 × 10^−28^	87.93%	YP_009191335.1
EFP6_16	9108	9881	R	negative regulator of beta-lactamase expression	Enterococcus phage vB_EfaP_IME195	0.0	95.72%	YP_009191336.1
EFP6_17	9882	10,274	R	Holin	Enterococcus phage vB_EfaP_IME195	2 × 10^−88^	99.23%	YP_009191337.1
EFP6_18	10,274	12,034	R	putative tail structural protein	Enterococcus phage vB_Efae230P-4	0.0	95.90%	YP_009103964.1
EFP6_19	12,036	12,824	R	hypothetical protein	Enterococcus phage vB_EfaP_IME195	1 × 10^−147^	78.24%	YP_009191339.1
EFP6_20	12,836	14,446	R	minor capsid protein	Enterococcus phage vB_EfaP_IME195	0.0	91.17%	YP_009191340.1
EFP6_21	14,415	15,056	R	lower collar protein	Enterococcus phage vB_EfaP_IME195	8 × 10^−124^	97.75%	YP_009191341.1
EFP6_22	15,013	16,050	R	collar protein	Enterococcus phage vB_EfaP_IME195	0.0	95.38%	YP_009191342.1
EFP6_23	16,067	17,278	R	putative major head protein	Enterococcus phage vB_Efae230P-4	0.0	95.26%	YP_009103959.1
EFP6_24	17,271	17,441	R	hypothetical protein	Enterococcus phage vB_Efae230P-4.2	6 × 10^−31^	96.43%	YP_009103958.1
EFP6_25	17,536	17,889	R	hypothetical protein	Enterococcus phage vB_EfaP_IME195	9 × 10^−14^	85.48%	YP_009191345.1

## Data Availability

The original contributions presented in the study are included in the article/Appendix A, further inquiries can be directed to the corresponding author.

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
