# Peer review of "Biological Traits and Comprehensive Genomic Analysis of Novel Enterococcus faecalis Bacteriophage EFP6"

_microorganisms, 2024, doi:10.3390/microorganisms12061202_

Round 1
Reviewer 1 Report
Comments and Suggestions for Authors
The manuscript entitled “Biological traits and comprehensive genomic Analysis of Novel Enterococcus faecalis bacteriophage EFP6” is well written and well structured.
The topic is addressed by the Authors in a very rigorous way and the experimental plan shows no gaps or inaccuracies.
I have to point out only small inaccuracies:
- in table 1 the name Klebsiella pneumoniae is written incorrectly because the final e of the diphthong is missing in the species
- Figures 2, 4, 5, 6 and 7 are too small and absolutely not readable and/or interpretable by the reader. It would be necessary to increase their size because their correct interpretation is essential for understanding the text
I cannot find any other corrections or suggestions to make to the Authors other than the extreme brevity of the conclusions which could also be expanded in light of the ideas discussed in a very interesting way in the "discussion" section.
Author Response
For research article
|
Response to Reviewer 1 Comments
|
||
|
1. Summary |
|
|
|
Thank you very much for taking the time to review our manuscript. Please find the detailed responses below and the corresponding revisions/corrections highlighted in red font in the re-submitted files.
|
||
|
2. Questions for General Evaluation |
Reviewer’s Evaluation |
Response and Revisions |
|
Does the introduction provide sufficient background and include all relevant references? |
Yes |
|
|
Are all the cited references relevant to the research? |
Yes |
|
|
Is the research design appropriate? |
Yes |
|
|
Are the methods adequately described? |
Yes |
|
|
Are the results clearly presented? |
Can be improved |
Revised |
|
Are the conclusions supported by the results? |
Can be improved |
Revised |
|
3. Point-by-point response to Comments and Suggestions for Authors |
||
|
Comments 1: In table 1 the name Klebsiella pneumoniae is written incorrectly because the final e of the diphthong is missing in the species
|
||
|
Response 1: Dear reviewer, Thank you very much for pointing this out, we have correct the spelling of Klebsiella pneumoniae this change can be found at page number 5, table 1, line number 213, serial number of bacteria in (table 1) 22 and 23, additionally we made a bit change in table 1 regarding its structure to delve a better look and accommodate less space in our manuscript. |
||
|
Comments 2: Figures 2, 4, 5, 6 and 7 are too small and absolutely not readable and/or interpretable by the reader. It would be necessary to increase their size because their correct interpretation is essential for understanding the text |
||
|
Response 2: Dear reviewer, Thank you for your valuable feedback regarding Figures 2, 4, 5, 6, and 7. We appreciate your comments and recognize the importance of these figures for the accurate interpretation of our study. We have replaced Figures 2, 4, 6, and 7 with enhanced quality images to ensure they are readable and interpretable. These updated figures are now included in the revised manuscript. However Figure 5 include text in small size therefore stretching them could violate the journal requirements set for figures therefore an increased sized and better quality image were moved to the supplementary files section and their respective informations were cited in the text as (Figure S2). this change can be found at page number 9, and line 275. |
||
|
4. I cannot find any other corrections or suggestions to make to the Authors other than the extreme brevity of the conclusions which could also be expanded in light of the ideas discussed in a very interesting way in the "discussion" section. |
||
|
Dear reviewer, Thank you for your constructive feedback on our manuscript. We are pleased to hear that you found the discussion section engaging and insightful. In response to your suggestion regarding the brevity of the conclusions, we have expanded this section to better reflect the key ideas and findings discussed in the manuscript. We believe this will provide a more comprehensive summary and reinforce the significance of our research. We appreciate your input, which has been invaluable in enhancing the clarity and impact of our conclusions.
|
||
|
5. Additional clarifications |
||
|
None |
||
Reviewer 2 Report
Comments and Suggestions for Authors
The manuscript describes a novel phage with activity against multidrug resistant virulant
E. faecalis strains.
The novelty is clearly established, but the authors should present the differences in the characteristics of the phage shown in this work in comparison to previous studies with other phages? Have the authors studied more or less characteristics than previous publications regarding other phages? Which are these characteristics? Please elaborate on this topic.
Comment about controls. Please describe clearly what controls did you use in this study. Please describe controls regarding laboratory material and regarding procedures. Also, did you use previously isolated phages as controls? If not, please justify; if yes, please describe.
Comment about figures. Some of the figures are low quality, hence they must be degigned from start in improved versions.
Comment about tables. These are OK, but possibly some may be transferred to supplementary material.
Comment about Discussion. Some parts of the Discussion should be moved in Results.
Comment about references. These are OK.
Comment about Concluding section. Please do not include new ideas in Conclusions. Please transfer these in Discussion. Also, please tone down Conclusions to bring the final word in line with the data presented in the manuscript.
Overall. Corrections as indicated and re-evaluation.
Author Response
For research article
|
Response to Reviewer 2 Comments
|
||
|
1. Summary |
|
|
|
Thank you very much for taking the time to review our manuscript. Please find the detailed responses below and the corresponding revisions/corrections highlighted in red font in the re-submitted files.
|
||
|
2. Questions for General Evaluation |
Reviewer’s Evaluation |
Response and Revisions |
|
Does the introduction provide sufficient background and include all relevant references? |
Yes |
|
|
Are all the cited references relevant to the research? |
Yes |
|
|
Is the research design appropriate? |
Must be improved |
Revised |
|
Are the methods adequately described? |
Must be improved |
Revised |
|
Are the results clearly presented? |
Can be improved |
Revised |
|
Are the conclusions supported by the results? |
Can be improved |
Revised |
|
3. Point-by-point response to Comments and Suggestions for Authors |
||
|
Comments 1: The novelty is clearly established, but the authors should present the differences in the characteristics of the phage shown in this work in comparison to previous studies with other phages? Have the authors studied more or less characteristics than previous publications regarding other phages? Which are these characteristics? Please elaborate on this topic.
|
||
|
Response 1: Dear reviewer, Thank you very much for pointing this out. We agree with this comment. Therefore we have added a comparison in the biological characteristics between our bacteriophage EFP6 and its closest bacteriophages E.phage Efae230p4 (NC025467.1) and phageN13(ON352054) in discussion page 15 line number 311 to 325, further more we reviewed a through literature against E. facealis bacteriophages most articles presents characteristics such as morphology, temperature, pH, Chloroform, determination of host range, additionally we found the 3 closely related bacteriophages with EFP6 were used for therapeutic potentials against their host strains. but it is difficult not to list these data for comparison with phages that have been isolated by other scholars. For completeness of the representation, we complement phages isolated from other scholars compared to these data supplemented phage-related comparisons isolated by other scholars, line 321-325. |
||
|
Comments 2: Comment about controls. Please describe clearly what controls did you use in this study. Please describe controls regarding laboratory material and regarding procedures. Also, did you use previously isolated phages as controls? If not, please justify; if yes, please describe. |
||
|
Response 2: Dear reviewer, To ensure the accuracy of results we conduct the experiments under sterile conditions in triplicate while carrying negative control to ensure and further support the accuracy of the results, however in chloroform experiment I mention the word control which refer to the phage EFP6 treating with no chloroform (0%). And we did not include any previously isolated or the phages shown in comparison with EFP6 as a positive control during laboratory procedures like temperature, pH e.t.c this is because other literatures does not include any positive control while conducting such experiments. |
||
|
Comments 3 Comment about figures. Some of the figures are low quality, hence they must be degigned from start in improved versions. |
||
|
Dear reviewer, Thank you for your valuable feedback regarding Figures. We appreciate your comments and recognize the importance of these figures for the accurate interpretation of our study. We have replaced Figures 2, 4, 6, and 7 with enhanced quality images to ensure they are readable and interpretable. These updated figures are now included in the revised manuscript. However Figure 5 include text in small size therefore stretching them could violate the journal requirements set for figures that is why an increased sized and better quality image were moved to the supplementary files section and their respective informations were cited in the text as (Figure S2). This change can be found at page number 9, and line 275. |
||
|
Comment 4 Comment about tables. These are OK, but possibly some may be transferred to supplementary material |
||
|
Dear reviewer, thank you so much for you suggestion we modify the structure of table 2 to delve a better look and accommodate less space in our manuscript |
||
|
Comment 5 Comment about Discussion. Some parts of the Discussion should be moved in Results. |
||
|
Dear reviewer, we revised the discussion section and we recognize that the data in the discussion section has been presented in the results, |
||
|
Comment about Concluding section. Please do not include new ideas in Conclusions. Please transfer these in Discussion. Also, please tone down Conclusions to bring the final word in line with the data presented in the manuscript. |
||
|
Dear reviewer thanks for the suggestion we revised the conclusion section and toned it down, and aligned the findings also we expand the conclusion a bit in light of discussion as suggested by the 1st reviewer. |
||
|
5. Additional clarifications |
||
|
none |
||